# Lessons Learned for Pre-Emptive Capture Management as a Tool for Wildlife Conservation during Oil Spills and Eradication Events

**DOI:** 10.3390/ani13050833

**Published:** 2023-02-24

**Authors:** B. Louise Chilvers, Pete J. McClelland

**Affiliations:** 1Wildbase, School of Veterinary Science, Massey University, Palmerston North 4442, New Zealand; 2Independent Researcher, 237 Kennington-Roslyn Bush Rd, RD2, Invercargill 9872, New Zealand

**Keywords:** pre-emptive capture, translocation, conservation, island eradication, oil spill, wildlife

## Abstract

**Simple Summary:**

Pre-emptive capture or translocation of wildlife during oil spills and prior to pest eradication poison applications have very specific conservation goals to ensure the survival of a threatened regional population or species. This paper reviews reports from pre-emptive captures and translocations of threatened wildlife undertaken during past oil spills and island pest eradications. Species captured, techniques used, outcomes of responses, and lessons learned were assessed and recommendations for the planning and trials needed for future pre-emptive capture operations are described. This paper aims to learn from the past to encourage better use and preparedness for pre-emptive capture as a preventative wildlife conservation tool in the future.

**Abstract:**

Pre-emptive capture or translocation of wildlife during oil spills and prior to pest eradication poison applications are very specific conservation goals within the field of conservation translocation/reintroduction. Protection of wildlife from contamination events occurs during either planned operations such as pest eradication poison applications, or unplanned events such as pollution or oil spills. The aim in both incidences is to protect at-risk wildlife species, ensuring the survival of a threatened regional population or entire species, by excluding wildlife from entering affected areas and therefore preventing impacts on the protected wildlife. If pre-emptive capture does not occur, wildlife may unintentionally be affected and could either die or will need capture, cleaning, and/or medical care and rehabilitation before being released back into a cleared environment. This paper reviews information from pre-emptive captures and translocations of threatened wildlife undertaken during past oil spills and island pest eradications, to assess criteria for species captured, techniques used, outcomes of responses, and lessons learned. From these case studies, the considerations and planning needs for pre-emptive capture are described and recommendations made to allow better use and preparedness for pre-emptive capture as a preventative wildlife conservation tool.

## 1. Introduction

As a wildlife management tool, the use of pre-emptive capture and translocations has risen rapidly in the past two decades [1,2,3]. In 2020, a review of 145 studies on wildlife capture and translocations indicated that 77% had been carried out for conservation purposes, predominantly to reintroduce or increase species’ presence within their indigenous range [3]. Here, we review pre-emptive capture and hold or translocation as techniques for preventing wildlife from entering contaminated areas or removing wildlife from areas before they are oiled or before the use of poisons for pest eradication purposes. The pre-emptive capture, holding, and translocation process undertaken for wildlife during oil spills and pest eradication poison applications is a specific conservation goal within the field of conservation translocation and reintroduction techniques [2]. The aim is to protect a significant proportion of a range-restricted species or significant regional population to reintroduce individuals back to their original range after an impact (oils spill or poison) has been removed to repopulate the area.

The first priority of wildlife protection during contamination events is to minimise the impact of the contaminant on wildlife through prevention. If a contamination event occurs, either planned (poisoning events for pest eradication) or unplanned (oil spills), the protection of wildlife can occur by: (1) stopping the contaminant from reaching and affecting non-target wildlife by containing poison in equipment that does not allow wildlife access or not spreading poison in critical wildlife areas, or containing the oil spill at or close to the source or stopping oil entering the wildlife habitat; (2) stopping wildlife being affected through the removal of wildlife from an affected or about to be affected area, or preventing wildlife from entering affected areas through hazing, deterrence, or pre-emptive capture. If these preventative measures are not undertaken, wildlife may die or need capture, cleaning, and/or medical treatment and rehabilitation before being released back into a clean environment. Avoiding wildlife from being impacted is always the highest priority as it prevents the duress, injury, and possible death of wildlife. Additionally, while it does have its own risks, it will significantly lessen the cost of a wildlife response if wildlife does become affected, and reduces the negative public and media reactions to reports and visual images of impacted wildlife [4]. This review concentrates on the priority of preventing wildlife from entering impacted areas or removing wildlife from areas before the area is impacted, and specifically focuses on pre-emptive capture either to translocate wildlife (move to another location) or to hold wildlife in captivity until release into a clean environment can occur.

The first step needed in all protective and preventative processes is developing a plan, based on analysis of areas at risk of impact (either oiling or where poison will be spread), the vulnerability of species (to both the contaminant and any proposed action), and potential response options for species at risk [5,6,7]. Each protective and preventative technique is species- and area-specific, and is usually initially based on a species population size and distribution, with species that have a high threat classification (i.e., listed as endangered), are range-restricted, and/or have high cultural importance/or public profile being most likely to be considered for pre-emptive capture. Threatened species usually have a small population size or restricted distribution or endemism, meaning an impact on their habitat could mean the extinction of that species or the local population. Other factors to consider are habitat use, therefore exposure risk, season, and biological factors, such as if the species is breeding at the time of impact [7]. For example, for pest eradication/poisoning events, how species forage is important, i.e., nectar-feeding birds are unlikely to be impacted by an aerial application of cereal poison bait; however, herbivores or omnivores may be vulnerable to primary poisoning as they could eat the bait directly, or omnivores, carnivores, or scavengers that could get secondary poisoning from scavenging poisoned individuals. In oil spills, any species that contacts, digests, or inhales fumes from oil can be affected, and, like poison operations, carnivores or scavengers can get secondary poisoning or oiling from predating or scavenging on other oiled wildlife. Undertaking pre-emptive capture of some species may not be practical or viable, i.e., large animals such as whales cannot be pre-emptively captured; therefore, hazing or deterrence are better options to undertake. For all species, the different stages in life cycles, such as breeding or moulting, can prevent other techniques such as deterrence or hazing from working effectively, and pre-emptive capture may be the only technique that could be successful. This was the case for New Zealand dotterels (*Charadrius obscurus*) in 2011 during the MV Rena spill in New Zealand as the dotterels were breeding when the oil spill occurred, making individuals very territorial, and animals would not have left their nesting sites, eggs, or chicks regardless if disturbance techniques were used.

This manuscript uses past oil spills and pest eradications using toxicants on islands as case studies of pre-emptive capture and holding, or translocation, to highlight lessons learned and considerations of species-specific response option restrictions, and outlines recommendations to allow better use, preparedness, and planning for pre-emptive capture as a conservation tool for threatened wildlife during contamination events.

## 2. Materials and Methods

An online literature search was undertaken aligned with the PRISMA 2020 guidelines [8] with the aim of creating a list of publicly available articles or reports on the use of pre-emptive capture during oil spill response or island eradication, from 1970 to 2022. Primary sources of information were sourced from scientific journal articles, conference proceedings, and any other grey literature through searches on Google, Google Scholar, or the Web of Science database (search terms were in English and included singular words or combinations of pre-emptive, pre-emptive, capture, wildlife, oil spill, oiled wildlife, and eradication). Additionally, searches were made through the Oil Spill conference websites for Interspill, IOSC, and translocation information from IUCN, including Proceedings of the International Conference on Eradication of Island Invasives. Experts in both fields were also contacted for any additional grey literature that was available but not yet published.

## 3. Results

The most striking result from this research is how few of the undertakings of pre-emptive capture of wildlife for prevention from contamination have been written into publicly available reports, journal articles, conference proceedings, or grey literature (Table 1). There have been over 600 island eradications of invasive rodents, many of which were multi-species eradications [9,10,11,12], and 1000s of oil spills that have affected wildlife [13]. There are multiple articles that highlight the need, advantages, and brief outlines on why pre-emptive capture should be undertaken but not many examples of when it has been undertaken or recommendations for what species should be considered, planning considerations needed, or factors to be taken into account before attempting pre-emptive capture [9,10,11,12,13,14]. However, even from the articles that do mention pre-emptive capture or translocation being undertaken, most only mention that it occurred, and there are few reports on how wildlife was captured or held, with what methodology, what proportion of the population was captured, processes during captivity, or short- or long-term survival or reproduction results after their release. Outlined below and in Table 1 are summaries of the 11 documented case studies of pre-emptive capture of wildlife during oil spill responses or island pest eradications that were assessed. Locations of case studies are shown in Figure 1.

### 3.1. Case Studies—Oil Spills

#### 3.1.1. Australia MV Iron Barron Oil Spill 1995

On 10 July 1995, the MV Iron Barron encountered bad weather coming into the port of Launceston in northern Tasmania, Australia, grounding on Low Head, Hebe Reef, leaking an estimated 325 tonnes of heavy bunker fuel oil [15]. Little blue penguins (*Eudyptula minor*) were significantly impacted by the spill with an estimated 10,000 to 20,000 killed and 1894 oiled birds captured, cleaned, and rehabilitated in an improvised rehabilitation facility [16]. The penguins were ready for release before their habitat had been cleaned, particularly as it was a large complex area with many islands over which the oil had spread. Rather than prolonging captivity, which increases the risk of disease and stress, and as breeding was imminent, a translocation strategy to release cleaned and rehabilitated penguins at different distances from the oiled site was trialled. This was undertaken to determine the optimal distance to release rehabilitated penguins so that they returned to their habitat after it had been cleaned. Twenty-five VHF-tagged penguins were translocated 360 km from the spill site on the east coast of Tasmania, and their movements were tracked from the air. Two birds returned to their original capture site within 3 days, not enough time to clean up the area, prompting a new release site 120 km further south (480 km in total). After the first trial, it was decided that the translocation site 480 km away was appropriate for the circumstances, and a further 863 penguins were translocated. At least 56% of the birds released further south returned to Low Head in four months, after their habitat had been cleaned. Monitoring found no differences in the survival rate of translocated and non-translocated birds.

Lessons learned: While translocation was considered effective in this situation, it is recommended that translocation protocols should be trialled before being implemented [15,16].

#### 3.1.2. South Africa MV Treasure Oil Spill 2000

The MV Treasure spilled approximately 400 tonnes of heavy fuel oil onto the coast of South Africa near Cape Town on 23 June 2000. The spill occurred near the two major breeding colonies, Robben and Dassen Islands, of the endangered African Penguins (*Spheniscus demersus*). A total of 19,000 oiled penguins were caught, cleaned, rehabilitated, and returned to a clean environment. Over 1660 birds died during captivity, most from the negative impacts of the oil [17]. To prevent even more penguins from being oiled, a further 19,506 penguins were captured, relocated, and released at Cape Recife near Port Elizabeth, ~700 km to the east of Cape Town [18]. These penguins, whether oiled or pre-emptively captured, represented over half of the known, endangered, declining population of African penguins at the time of the spill [19].

Relocated birds returned quickly to their breeding islands, with the faster returning in 11 days and most returned within two to four weeks [17]. This indicated that Cape Recife was an appropriate location for release because it was a suitable distance to allow time for the oil to be cleared before the birds returned, but close enough for birds to return within a month, thereby minimising any disruption to breeding and moulting. Of the 19,506 penguins translocated, 241 died between being captured and release at Cape Recife due to some being transported in closed trucks causing CO_2_ poisoning. Additionally, before transport, those kept on Dassen Island were kept fenced in an area on the island with limited access to drinking water and no areas to swim. Both factors contributed to the higher mortality of those pre-emptive captured birds [19]. Additional to the adults, 3350 orphaned chicks were also pre-emptively captured and reared in captivity and released back into their clean environment when they had fledged. Of the 3350 chicks collected, approximately 2300 were fledged and released [17].

Prior to the MV Treasure spill, South Africans’ seabird oil spill rescue plans focused on catching and treating oiled birds as soon as possible, before releasing them back into the wild; preventing birds from becoming oiled was not part of any plan [19]. This wildlife response is still the largest relocation response for oiled wildlife globally and, due to its success, the implementation of relocating birds before they became oiled has been implemented as a response option and documented to have been an effective conservation measure [17]. One year after the MV Treasure spill, 84% of the evacuated birds had been re-sighted, compared with 55% of the captured, cleaned, and released birds.

Lessons learned: The two overall lessons from the pre-emptive capture of African penguins were greater consideration of conditions prior to and during transport to translocation sites to prevent deaths, and consideration of distance transported so that the wildlife’s return allowed enough time for the oiled areas to be cleaned, but the distance was not too far to cause individuals to get disorientated or lost, or to cause major disruption to breeding or moulting cycles. A second conclusion is that pre-emptive capture and raising of penguin chicks is a successful conservation practice that continues today for African Penguins, not only during oil spills, but also droughts, colony disturbances, and other human and natural impacts on this endangered species (https://sanccob.co.za; accessed on 15 February 2023).

#### 3.1.3. USA Deepwater Horizons Oil Spill 2010

On 20 April 2010, the Deepwater Horizon well exploded 66 km off the coast of Louisiana, in the Gulf of Mexico, and before being capped, three months later, more than 780,000 tonnes of crude oil were spilled. There were numerous impacts on the environment and wildlife, and because of the length of time oil continued to be spilled, some wildlife that had been cleaned and rehabilitated were ready for release long before their environment was cleaned. Brown Pelicans (*Pelecanus occidentalis)* were one of the species impacted, with more than 700 rehabilitated in south-eastern Louisiana alone [24]. To overcome the lack of a clean habitat for their release, 182 oil-rehabilitated pelicans were translocated from south-eastern Louisiana to Rabbit Island in south-western Louisiana, an island that was not impacted by the spill and had non-impacted pelicans breeding on it. The aims of this translocation were to enable monitoring of movements of translocated groups and determine if translocation would delay pelicans returning to their habitat, and therefore getting re-oiled, and to be able to monitor mortality, determine the integration of translocated pelicans with local pelican groups, and determine if supplemental feeding of translocated birds prolonged occupation on the island, therefore again reducing the likelihood of re-oiling [20]. Daily surveys were undertaken at the island for six weeks from the date of translocations, with supplementary feeding occurring twice a day for four weeks. There was no mortality of rehabilitated birds recorded and it was observed that translocated pelicans mixed readily with local pelican flocks. Many of the local and translocated pelicans moved away from the island within 4 to 6 weeks, likely due to natural and human-induced factors.

Lessons learned: The translocations and supplementary feeding program of the brown pelican were considered successful at reducing the movement of pelicans back into oiled areas. However, habituation to the feeding vessel and supplementary feeding were observed both from the rehabilitated and local pelicans. For future translocations, it is suggested shorter time periods of supplemental feedings should occur, using alternative feeding strategies such as blinds or remote feeders due to the easy habituation of pelicans to humans. For tracking of movement of rehabilitated birds, it is recommended a subset of individuals be radio/satellite tagged for documentation of movements and mortality.

#### 3.1.4. New Zealand MV Rena Oil Spill 2011

On 5 October 2011, the container vessel MV Rena ran aground on Astrolabe Reef, Bay of Plenty, New Zealand, and within days spilled approximately 350 tonnes of heavy fuel oil. The endangered Northern New Zealand dotterels (*Charadrius obscurus aquilonius*—a small ~140 g shorebird) were pre-emptively captured as part of the oiled wildlife response to ensure the survival of a regional population. The pre-emptive capture occurred as it was considered that if these small birds became significantly oiled their chances of survival were minimal despite cleaning/rehabilitation [21]. Sixty dotterels were caught, with over half the birds already having some level of oil contamination. This population of dotterels represented ~6% of the global population of this species and the majority of the local population within the area of the spill. Many pairs were already breeding and nesting at the time of the spill, so other deterrence or hazing activities would not have worked as the birds are territorial and would not have moved away from their nests. This was the first time wild adult New Zealand dotterels pairs had been held in captivity for a prolonged period. Birds were caught in their breeding pairs, with each pair held in individual enclosures, blocked from view of other pairs to prevent territorial and fighting behaviours which would have been normal during breeding. There was a 90% survival rate of the New Zealand dotterels held in captivity during the MV Rena oil spill response over a ~2-month period [21]. Dotterels took 1–15 days (median 5 days) to convert to the captive diet. Sixty-one percent of birds obtained minor abrasions from contact with enclosure netting during captivity due to their flighty behaviour which did not affect survival; however, seven birds (11.7%) developed respiratory disease, with six of these dying from aspergillosis causing pneumonia-type deaths [21]. Intensive captive husbandry was needed to convert the birds to a captive diet, minimise injuries, and manage pododermatitis/foot sores.

Lessons learned: It was critical to have a dedicated captive management team for these birds. The challenges that come with managing wild adult shorebirds in captivity and converting to captive diets are well recognised within the wildlife rehabilitation community. Additionally, shorebirds are species considered to respond poorly to the stresses of capture and captivity [35]. Therefore, although the pre-emptive capture and management of shorebirds during an oil spill to minimise the effects of oil spills carries significant costs and risks to the birds, it is considered essential in emergency management situations for high-priority/at-risk species. Additional to normal capture stressors, clinical signs of respiratory disease were not observed until the last half of the time the birds were in captivity. Therefore, a strong recommendation for the management of shorebirds that are pre-emptively captured is that the clean-up of their habitat is prioritised to enable the early return of birds to the wild.

### 3.2. Case Studies—Island Eradications

#### 3.2.1. New Zealand—Mice and Rat Eradication/Poisoning, Kapiti Island 1996

After the eradication of cats (*Felis catus*), deer (*Cervidae* spp.), pigs (*Sus* spp.), goats (*Capra* spp.), and possums (*Trichosurus vulpecula*), by hunting and trapping, from the rugged 19.65 km^2^ Kapiti Island off the south-west coast of the North Island, New Zealand, the Department of Conservation of New Zealand also successfully eradicated Norway and Pacific rats (*Rattus norvegicus* and *R. exulans*) in 1996 using helicopter broadcast of brodifacoum cereal baits [22]. Trials with non-toxic baits were carried out on North Island weka (*Gallirallus australis grey*) and little spotted kiwi (*Apteryx owenii*), both flightless birds found on the island, to help determine the risks of poisoning for these non-target species [27]. North Island weka at the time were classified as endangered and were expected to be affected by the eradication activities both from primary and secondary poisoning, particularly as weka are omnivores and will scavenge and kill other species. From the non-toxic trials, measures to minimise the effects of the poison application on fauna at risk were put in place, which included the capture and holding in captivity or translocation to reserves on mainland New Zealand of 243 weka, and the transfer of 66 New Zealand robins (*Petroica australis*), which had previously been identified as being at risk to nearby Mana Island. Post-poisoning call rate monitoring indicated that weka call rates were significantly lower after poisoning; however, it could not be determined if that was caused by the removal of weka from the island (not yet returned or released at the time of the call counts) and/or the poisoning operation, because no call rate monitoring was undertaken in the period between the removal and the poisoning for comparison. However, the fact that weka calls were heard meant that some survived the poisoning operation, and together with the birds released after the operation, they are now distributed throughout Kapiti Island and breeding prolifically [22].

Lessons learned: Species at risk should be identified through both non-toxic bait trials and knowledge from species at risk from previous operations. Monitoring between pre-emptive capture and poison applications should be undertaken to allow the determination of the impacts of both.

#### 3.2.2. New Zealand—Rat Eradication/Poisoning, Whenua Hou Nature Reserve/Codfish Island 1998

Whenua Hou Nature Reserve/Codfish Island is located 3 km NW of Stewart Island, New Zealand, and is the protected island home to the largest population of the endangered Kākāpō (*Strigops habroptilus),* a large flightless native parrot. Following the removal of possums and South Island weka (*Gallirallus australis australia)*, eradication of the Pacific rat was undertaken on Codfish in August 1998, using a combination of aerial applications and bait station cereal pellets containing brodifacoum. In preparation for the eradication, a smaller island, Putauhinu (96 ha), was eradicated of Pacific rats the year before in 1997, so that a population of fernbirds (*Bowdleria punctata wilsoni*), endemic to Codfish Island, could be established [23]. Additional to the transfer, a 37 ha block of the best fernbird habitat known on Codfish, containing the densest population of fernbirds, was poisoned using bait stations at 25 m intervals instead of using aerial baiting, which had been shown during field trials elsewhere to cause a high death rate in fernbirds. All Kākāpō (except one that could not be found) were removed from the island prior to the poison application and temporarily held on a separate island. Short-tailed bats (*Mystacina tuberculatus tuberculatus*) were also managed, with 50 being captured and released onto Ulva Island, a predator-free island off Stewart Island; however, this was unsuccessful. Additionally, during the poison applications, four purpose-designed “batteries” were constructed on Codfish Island with 386 short-tailed bats held for nearly three months. There was no observable loss to the bat population linked with the bait application although individuals are likely to have been lost. Nine bats were lost up until the last week of the capture program, when 42 died in one event due to heat stress in one of the roost boxes. Despite this sad event, the operation was still considered a success given how difficult bat husbandry can be. The bat protection and monitoring was undertaken by a team of 5–7 people and this investment of single-task personnel is one of the main reasons for its success. The 21 fernbirds that were transferred to Putauhinu were confirmed to have bred, and follow-up checks on Putauhinu have shown that the fernbird population has continued to increase and expand its range on Putauhinu. It appeared most fernbirds were lost on Codfish due to the bait application, despite the management, with very few recorded for 2 years after. However, enough survived to rebuild and recover not only to the population’s original range, but to also expand into a variety of habitats in the absence of rats [23]. This meant the planned reintroduction from Putauhinu was not required.

Lessons learned: Dedicated husbandry teams are needed for the pre-emptive capture of species during eradication projects. Although the bait stations in the areas of the fernbirds achieved the goal of reducing fernbird mortality, it was thought that as fernbirds outside the area affected by the aerial bait died, fernbirds within the bait station area expanded their range and therefore became more exposed to aerially laid bait. Therefore, it was thought that the impact may have been lessened by expanding the size of the core area in which only bait stations were laid, thus increasing the percentage of birds within the core area. This result also led to the conclusion that the additional cost of rat eradication and transfer of a security population to Putauhinu was warranted even though it proved to not be necessary. This eradication also proved that field trials are important for poison eradication, as fernbirds were thought to be insectivores mainly preying on spiders and hence at little risk from the baiting operation. However, field trials showed that fernbirds when presented with brodifacoum bait would eat it, and indicated that the species would be heavily impacted by aerial bait, therefore leading to the mitigation work of bait stations in the area where the fernbirds were in high abundance (Pete McClelland pers comm).

#### 3.2.3. Seychelles—Cat, Rabbit, Rats and Mice Eradication 1996–2000

Between 1996 and 2000, attempts were made to eradicate five introduced mammal species, feral cat, rabbit (*Oryctolagus cuniculus*), ship rat (*Rattus rattus*), Norway rat, and house mouse (*Mus domesticus*), on four inhabited Seychelle islands. As there were no rat-free islands in close proximity for the transfer of species at risk, 590 individuals from three threatened native species, the Seychelles magpie-robins (*Copsychus sechellarum*, *n* = 39), Seychelles fodys (*Foudia sechellarum, n* = 330, 50% of the known population), and the Aldabran giant tortoises (*Geochelone gigantea, n* = 218), thought to be at risk from primary and/or secondary poisoning, or for public goodwill in the case of tortoises, were held in captivity for the three months of the eradication program [24]. During the captivity of these species across the islands, the avicultural knowledge and capability of staff increased enormously. The captivity of these species during eradication was very successful, with magpie-robins breeding successfully during three months in captivity [25]. All tortoises, Seychelles fodys, and magpie-robins were successfully released within 3 months after bait application.

Lessons learned: Dedicated husbandry teams are essential for success and allow for increased knowledge and capability for the aviculture of species and in the region. This was one of the first major human-occupied island eradication programs and its success led to the planning of eradications on the likes of Galapagos and Lord Howe Islands (see below). It was an important conclusion at the time that land held privately, human habitation, or tourism activities need not be seen as barriers to eradication projects, as island-based tourism activities can provide the financial and human resources to restore and maintain threatened endemic biodiversity.

#### 3.2.4. California, USA—Rat Eradication/Poisoning, Anacapa Islands 2001–2002

Eradication of black/ship rats from Anacapa Islands, US Channel Islands National Park, California, was undertaken in 2001 and 2002 [26]. This was the first aerial application of a rodenticide in North America and the first attempt in the world to eradicate a rodent from islands while preserving a native endemic rodent on the same islands. There are three islands in this group and, to ensure the presence of the native deer mouse (*Peromyscus maniculatus anacapae)*, the rodenticide application was staggered over two years so that a wild population was always present on one or more islands [26]. Concurrently, mice populations from each island were held in captivity during poison applications. Additional to the mice, to avoid as much as possible birds being affected by the application, bait was made using colouring and sizing that deterred gulls and granivorous birds, resident raptors were captured and held or translocated, and a 15 ha no-drop zone was established on West Anacapa to create a refuge for granivorous birds, particularly the Santa Cruz Island rufous-crowned sparrow *Aimophila ruficeps obscura.* In the no-drop zone, rats were poisoned using bait stations that were inaccessible to granivorous birds.

Prior to the first poisoning in 2001, 185 deer mice were live captured from East Anacapa and held for five months. Of these, 174 were released after poisoning. Prior to the second poisoning in 2002, 373 and 365 deer mice were captured from Middle and West Anacapa, respectively, and held in captivity, while concurrently 715 and 308 mice from Middle and West Anacapa were captured and translocated to rat-free East Anacapa. Five months after the second eradication, 358 and 360 captive mice were reintroduced to Middle and West Anacapa, respectively. Raptors were live captured prior to rodenticide applications (including eight peregrine falcons *Falco peregrinus*, nine red-tailed hawks *Buteo jamaicensis*, four barn owls *Tyto alba*, and six burrowing owls *Athene cunicularia*). Most were released onto suitable habitat on mainland California, except peregrine falcons, which were held and released back onto Anacapa 3 weeks after rodenticide applications. A total of 94 birds (16 species) were identified from carcass searches following rodenticide applications. Of the 63 birds tested for brodifacoum, 59 (94%) tested positive [26].

Lessons learned: The successful recovery of the Anacapa deer mouse following the eradication demonstrates that it is feasible to eradicate invasive rodents from islands when native rodents or other susceptible native animals can be held in captivity and kept away from poison. Captive holding and translocation significantly reduced raptor mortality.

However, captive holding or other mitigation measures (no-drop zones) may be necessary for sedentary granivorous passerines, as previously used for fernbirds during the eradication of rats from Codfish Island.

#### 3.2.5. Galapagos—Rat Eradication/Poisoning, Pinzón Island 2012

In December 2012, brodifacoum bait was spread on Pinzón Island (1815 hectares), Galapagos, to eradicate black rats which had prevented the Pinzón giant tortoise (*Chelonoidis duncanensis*) from breeding successfully for nearly a century. Two years prior to the poisoning, 15 adult Pinzón tortoises were brought into captivity and housed on Santa Cruz Island for release after the eradication; all survived, and breeding has been recorded since the eradication. The two other species of concern were Pinzón lava lizards (*Microlophus duncanensis*) and Galapagos hawks (*Buteo galapagoensis*). Forty Pinzón lava lizards were taken into captivity prior to baiting and maintained in enclosures on Pinzón Island, and were released 10 days after the second bait application as it was determined that, due to bait degradation, the risk of poisoning would by then be minimal [32]. Two lava lizards escaped captivity and five captive lizards died during captivity, resulting in a survival rate of 87%. Sixty Galapagos hawks were taken into captivity and held in purpose-built aviaries on Pinzón Island. All survived captivity and were released 12–14 days after the second aerial bait application. However, within 12 to 170 days after release, 22 mortalities of tracked Galapagos hawks were recorded [27]. Unfortunately, reported to be due to the arid conditions of the island, residual poison persisted in lava lizards. The remaining Pinzón Island Galapagos hawk population (*n* = 10) was recaptured, returned to captivity, and treated with Vitamin K1, while the toxicological levels of Pinzón lava lizards were monitored [27]. These captive Galapagos hawks represented 15% of the original population and were released when risk was considered acceptable, in July and August 2016. As of 2018, eight hawk nests had been observed on Pinzón with chicks and fledglings confirmed.

Lessons learned: The rodenticide used in this eradication remained in the ecosystem much longer than in any previous rodent eradication project worldwide. This resulted in the secondary poisoning of predatory hawks long after expected; therefore, understanding the longevity of poisons in the local environment and possible pathways into at-risk species is essential to ensure captive wildlife are held for an appropriate length of time so as not to be impacted. Similar to the Whenua Hou fernbird experience, this eradication also highlighted the importance of field base trials, as laboratory trials do not always reflect the response of wildlife in the field. Lava lizards did not eat the rodent bait in the laboratory; however, they did in the field, leading to a greater impact on themselves and the hawks than expected.

#### 3.2.6. Australia—Rat Eradication/Poisoning, Lord Howe Island 2019

Lord Howe is a permanently human-inhabited island group approximately 1455 hectares in size and having a diverse landscape, where rats have already been implicated in the extinction of five endemic bird species and at least 13 species of endemic invertebrates. After the successful eradication of cats, pigs, and feral goats from the Lord Howe Island group, ship rats and mice were then targeted. In 2019, brodifacoum baits were distributed across the island depending on habitat type and land use using aerial distribution in the uninhabited areas, and hand broadcast and locked bait stations in the inhabited areas. Following field observations on a range of species on the island in which they were presented with non-toxic bait, two species were thought to be at risk from the bait, Lord Howe woodhen (*Gallirallus sylvestris)* and Lord Howe pied currawong (*Strepera graculina crissalis*), and successful pre-emptive captive trials were undertaken for these species in 2013, prior to baiting. Twenty-two woodhens and ten currawongs were captured and held in captivity, with all individuals subsequently released successfully back into the wild. The woodhens were captured in family groups or pairs and held together in pens, and initial trials showed the need to be careful with the species’ diet as they put on weight quickly on the captive diet [28,29]. From this trial, despite the woodhens normally being very territorial, they were held in groups of 20–30 with great success. The idea to hold the woodhens together was undertaken from the experience with weka (a similar bird to the woodhen), on Kapiti Island, New Zealand (see above). For the main poison application, to minimise any potential impact, at least 85% of the woodhen population and 50% of the pied currawong population were placed into captivity. Birds were held for at least one month before baiting, and until risks of primary or secondary poisoning were considered no longer present. From ongoing surveys of the island, by the second autumn woodhen survey following the rodent eradication, 778 woodhens were recorded over a two-week period. This number nearly quadruples the population survey results prior to rodent eradication.

Lessons learned: The Lord Howe Island eradication showed how important it was to build on the learning from previous operations based on similar species, i.e., woodhens vs. weka, not only to decide which species need managing but how they can be managed. The undertaking of pre-emptive capture trials before the poison application allowed a greater understanding of how animals would react to captivity, including understanding that they can put on weight easily with captive diets and can be held together in larger numbers than normal when needed, and assuring the local community of its success. This understanding allowed for better-conditioned individuals to be released back into their environment, with current surveys showing woodhen are thriving on the rodent-free island.

#### 3.2.7. United Kingdom/South Atlantic—Mouse Eradication, Gough Island 2021

An attempt was made by the Royal Society for the Protection of Birds (RSPB) and Tristan da Cunha to eradicate mice from the rugged 6500 ha Gough Island between June and August 2021. Gough Island is part of a World Heritage Site in the southern Atlantic and is one of the world’s most important seabird breeding areas, with 22 species of seabird species breeding on the island, many of which are globally threatened, as well as two endemic threatened land bird species. Invasive non-native mice have been responsible for demographically unsustainable levels of chick mortality in seabirds [30,31]. However, it was the two endemic land birds, the Gough bunting (*Rowettia goughensis*) and Gough moorhen (*Gallinula comeri*), for which primary and secondary poisoning was of greatest concern during the eradication, as many of the seabirds would not be present on the island at the time of the poison application. Trials on the capture and holding of these two land bird species began early in the programme planning with 25 buntings and 30 moorhens captured and held for 6 weeks between April and September 2010. Over March to May 2021, 84 moorhens (pre-eradication population estimate of 3500–4250 pairs) [32] and 100 buntings (from a population of 1041–1889 individuals; RSPB unpublished data) were captured and held in captivity during the mouse eradication poison application. Eighty moorhens and 103 buntings were subsequently released back into the wild after the completion of the bait application. Follow-up surveys had shown that there was still a significant wild population of buntings after the bait applications that was then joined by the safeguard bunting population. However, as expected, the wild moorhen population was significantly impacted, hence the importance of the aviculture operation. In late September 2021, once any sign of bait on the island and especially in the lowland moorhen habitat had disappeared (extensive searches were undertaken to validate this), the captive moorhen population was released into their preferred habitat. At the time of writing the status of the moorhen population is still not known: monitoring has proved difficult (e.g., few birds calling, larger potential habitat available, dense vegetation), and while moorhens remain on Gough, no breeding has as yet been recorded to show that the population is beginning to rebuild (A. Callender pers comm.).

Lessons learned: It is highly recommended that the avicultural project be run as a separate parallel operation so as not to be overshadowed in its importance by the “high-profile” baiting operation. A dedicated husbandry team was essential for the capture, care, and survival of these two species. A comprehensive plan for all stages of the aviculture operation is required and it should be followed unless there is good justification to do otherwise. Adequate resourcing is crucial, especially considering sufficient capacity over the holding period to allow members of the team to have downtime, particularly on remote islands such as Gough. The documentation and recommendations from the pre-poisoning capture and holding of moorhens and buntings from Gough Island was an important tactic in the success of the survival of the species for release after the eradication attempt [33]. Trials need to be carried out early and critically accessed, and final design undertaken by a combination of aviculturists and eradication specialists where appropriate so, if necessary, teams can work together. Trials should aim to hold the birds for as long as they are likely to be held for the operation when possible, as the initial Gough trials were not of sufficient length to test how issues such as pododermatitis might affect the birds.

Unfortunately, the eradication attempt was not successful, possibly due to slug consumption of the aerial spread bait, which reduced the amount available for rodents and meant not all mice accessed a lethal dose [34]. An independent review panel is currently assessing the Gough eradication attempt and will report its findings in 2023.

## 4. Discussion

The conservation goal of pre-emptive capture/translocation of threatened wildlife during oil spills or eradication operations is to protect a biologically significant proportion of a range-restricted species or significant regional population to reintroduce individuals back to their original range to re-establish the population after an impact has been removed. There are examples both in oiled wildlife response and, particularly, eradications where wildlife was considered to be at risk, however, the at-risk population was not a biologically significant proportion of the species, range-restricted species, or significant regional population (i.e., could be reintroduced from other regions if impacted). Examples of these include eradications that impacted Giant Petrels (*Procellariiformes* spp.) and Skuas (*Stercorariidae* spp.) on Macquarie Island, Australian subantarctic [36], Antipodes Island, New Zealand subantarctic [37], and South Georgia, South Atlantic [38]. It should also be noted that these species would also be difficult to hold in captivity, and particularly to hold sufficient numbers of individuals for the risk period to re-establish the population.

The main result from this review is how few of the pre-emptive capture/translocations have occurred for oiled wildlife response and how few have been documented for island eradication operations considering the number of both that have occurred in the last three decades [10,11,12,13]. Despite this, there are still valuable lessons to be learned from what has been documented.

The most important lessons learned from both responses is the importance of planning and a specific, dedicated team for the capture and care of captive wildlife. In the case of eradications, planning can be very specific as the site and species present are known well in advance. Whereas planning for oiled spills is more likely to be generic because the specifics of the event, e.g., timing, location, season, etc., are uncertain. However, there will be known endangered or range-restricted species that can be identified within a region or country that can have plans developed for them in advance in the case of a spill in their area. In general, eradication operations have significantly more time and ability to learn from both laboratory and field-based trials, including non-toxic bait trials, to determine species likely to be at risk, and to be able to trial the capture and holding of species prior to poisoning event, as seen for Gough and Lord Howe [28,33]. However, it is only recently that the capture and captive care of protected wildlife has been undertaken by specialist rehabilitators/zoological carers (Lord Howe, Taronga Zoo, and Gough, Royal Society for the Protection of Birds) for eradication operations, and that there has been greater documentation and reporting of the methodology of pre-emptive capture and care techniques, successes, and recommendations.

Conversely, since the Exxon Valdez oil spill in 1989, oiled wildlife response has almost always been undertaken by professional and/or experienced wildlife veterinarians or rehabilitation centres, and has involved greater monitoring and documentation of events. Unfortunately for wildlife, due to the random, unexpected, and usually instantaneous nature of oil spills, and lack of planning, pre-emptive capture has not been undertaken frequently in oiled wildlife response. The speed at which the oil covers and impacts the environment and wildlife is often too fast to allow pre-emptive activities to occur; however, those that have occurred have been reasonably documented (i.e., [15,16,17,19,21,39]). Additionally, this review has highlighted that it is not only pre-emptive capture and translocation prior to wildlife being oiled that is a successful, useful management tool for oiled wildlife response. The translocation of cleaned rehabilitated birds outside the area of the oil, to lengthen the time before wildlife return to their habitat, has been shown not only to reduce their chance of being re-oiled, but also reduced the time spent in captivity, therefore reducing secondary problems that can occur in captivity such as pododermatitis [21].

For both oil spills and eradications, an essential lesson is understanding and using the knowledge learned from previous operations to improve current operations. These learnings can be everything from understanding species likely to be impacted, to how to manage and care for species, and for how long to keep them in captivity. However, this type of information needs to be written up and made publicly available from past responses for these lessons to be learned and used in the future.

All reports and articles on pre-emptive capture outline how logistically challenging it can be and, depending on preparation time prior to an oil spill or eradication poison application, pre-emptive capture can seem almost unfeasible. However, the case studies above outline how, with consideration and planning, particularly for planned eradication poison applications, pre-emptive capture or translocations can be successful and save significant proportions of populations or range-restricted species from potential extinction events. For an oil spill event, decisions for pre-emptive capture must be made in a time-critical window, meaning delays could result in wildlife being oiled or injured, or dying. Therefore, understanding the requirements as suggested below prior to a spill occurring, and therefore the activation of a predetermined wildlife plan and personnel immediately after an oil spill, is needed to ensure protective and preventive actions can be undertaken if the situation allows [13].

Primary requirements for pre-emptive captures include:Before a spill or eradication—determine potential species at risk: consider the numbers and species of wildlife, their threat classification and geographic extent, the animals’ behaviour (seasonal, feeding, breeding), response options available for each species, and whether it is practical for the species to be kept in captivity or if capture and translocation are more appropriate [13]. For eradication activities, this includes developing an inventory of non-target species, including bait-competitors, and a simple food web model to try and understand all possible primary and secondary poison routes pathways (e.g., [40]). Both laboratory and field trials are recommended.Conduct applicable capture planning (techniques and personnel) to ensure animal welfare, i.e., conduct site assessment for capture and housing: consider site accessibility and the prioritisation of locations (accessibility, tide, weather), and have knowledge of species behaviour and the geographical area, and lists of experts and pertinent contacts.Plan for appropriate captive care arrangements (housing, husbandry, personnel expertise, etc.).Plan and possibly trial relocation solutions (release location, transport, site fidelity, predicted time to return, energetic costs of return, etc.).Ensure the plans for aviculture can logistically be undertaken given the species, scale of operation, and numbers of individuals or species that need to be held.Critically, gain approvals from relevant government agencies and first nations groups, where applicable, for the capture, handling, and holding and transfer/release of wildlife.

## 5. Conclusions

The difficulty of capturing wildlife safely and providing for their health in a captive environment or during relocation must never be underestimated. The risk of impacts from oiling or primary or secondary poisoning must be weighed against the risks of injury, disease, or death of wildlife during the entire pre-emptive capture and holding process.

To determine the effectiveness of a wildlife pre-emptive capture process, it is critical to monitor and quantify the short- and long-term success or failure of the project. Relative to the number of oil spills and eradication operations that have occurred, direct counts of mortality and pre- and post-event wildlife monitoring studies are still rare. These types of research are critical to fully understand the total and long-term benefits of pre-emptive capture operations of wildlife [41,42,43]. One of the strongest recommendations from this review is that, once species are identified that are suitable and likely to require pre-emptive capture and holding or translocation, the development of prospective techniques for them should be undertaken by a dedicated and experienced team, and fully documented, with outcomes made publicly available to inform future conservation planning.

## Figures and Tables

**Figure 1 animals-13-00833-f001:**
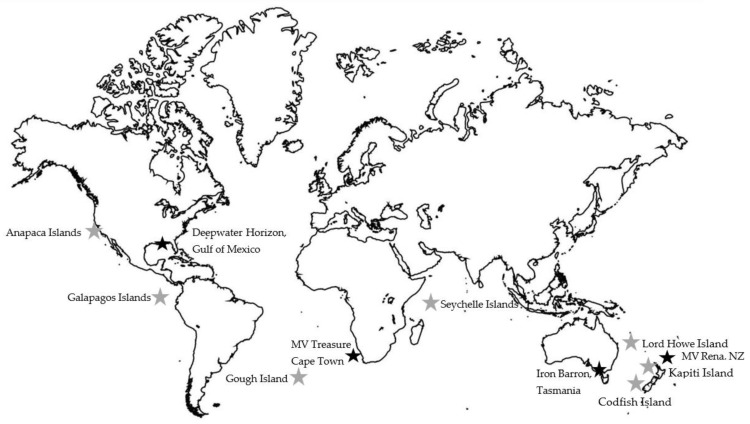
Locations of case studies. Oiled wildlife responses are shown as a black star. Eradication operations are shown as a grey star.

**Table 1 animals-13-00833-t001:** Case study summaries highlighting species protected, control measures, outcome and lessons.

Case Study (Reference)	Species Protected	Control	Outcome	Lessons
Oil Spill Response				
MV Iron Barron, Tasmaina, 1995 [15,16]	Little blue penguins (*Eudyptula minor*)	Translocation 480 km from spill after wildlife cleaned and rehabilitated to allow time for area to be cleaned before wildlife returns	863 translocated, 56% reported returned within 4 months, no difference in survival rates recorded between translocated and non-translocated wildlife	Translocation considered effective, recommend trialing distances before being implemented
MV Treasure, South Africa, 2000 [17,18,19]	African Penguins (*Spheniscus demersus*)	19,506 penguins were pre-emptive captured and translocated ~700 km away to allow time for area to be cleaned before wildlife returns. 3350 orphaned chicks captured and hand reared	One year after the spill, 84% of the translocated birds were re-sighted, compared with 55% of the captured, cleaned, and released birds. Of the 3350 chicks collected approximately 2300 were fledged and released	Translocations considered effective however greater consideration of conditions prior to and during transport needed. Preemptive capture and hand rearing of chicks was a successful conversation practice which can be used for oil spills, droughts and other human and natural impacts.
Deepwater Horizons, USA 2010 [20]	Brown Pelicans (*Pelecanus occidentalis)*	Translocation and supplementary feeding away from spill area after wildlife cleaned and rehabilitated to allow time for area to be cleaned	No morality of translocated birds reported, birds mixed with local flock and stayed for 4 to 6 weeks	Translocations and supplementary feeding considered successful. Shorter time period of feeding suggested and tracking of translocated individuals
MV Rena, New Zealand, 2011 [21]	Northern New Zealand dotterels (*Charadrius obscurus aquilonius)*	60 dotterels pre-emptively caught and held for 60 days	90% survival to release	Critical to have a dedicated captive management team. Strong recommendation that if shorebirds are preemptively captured, that the clean-up of their habitat is prioritised to enable as early release as possible.
Eradication Operation				
Kapiti Island, New Zealand 1996 [22]	North Island weka (*Gallirallus australis grey)*	Capture and transfer of 243 weka to mainland NZ	Some Weka not transferred survived the aerial poisoning and no reintroduction back to the island was made. Weka now breed prolifically on the island and are fully recovered	Species at risk should be identified through both non-toxic bait trials and knowledge from species at risk from previous operations
Whenua Hou/Codfish Island 1998 [23]	Fernbirds (*Bowdleria punctata wilsoni*), Short-tailed bats (*Mystacina tuberculatus tuberculatus*)	Fernbirds—21 birds transferred to a nearby rat-free island and poison placed in bait stations in highest density fernbird habitat instead of aerial spread Bats—captured and translocated onto another island and 386 held in captivity on island for ~90 days	Fernbirds—transferred birds successfully translocated, established, and bred and have not been transferred back. Most fernbirds on the island were thought to be killed. However enough survived or naturally reintroduced to recover and expanded their range without rats. Bats—capture and release unsuccessful, none know to survive. Capture and hold on the island was considered successful	Dedicated husbandry teams are needed for the pre-emptive capture of species during eradication projects. The additional cost of an additional rat eradication and transfer of a security population to another island was considered warranted even though not needed in the end.
Seychelles 2000 [24,25]	Seychelles magpie-robins (*Copsychus sechellarum*), Seychelles fodys (*Foudia sechellarum),* Aldabran giant tortoises (*Geochelone gigantea)*	590 individuals from the 3 species were held in captivity on the island for up to 90 days during eradication	All individuals survived capture and were released. Magpie robins breed in captivity	Dedicated husbandry teams are essential for success and allow for increased knowledge and capability for the aviculture of species
Anacapa Islands, California 2001 and 2002 [26]	Anacapa deer mouse (*Peromyscus maniculatus anacapae*), Peregrine falcons *(Falco peregrinus)*	Aerial poisoning was conducted over two years. Pior to each drop deer mice were live captured and held in captivity or before the second application mice (from the soon to be poisons island) were transferred into the wild on the now rat-free island Raptors were live captured prior to rodenticide applications (peregrine falcons, red-tailed hawks, barn owls, and burrowing owls). Most were released on the mainland in suitable habitat; peregrine falcons were held and released back onto Anacapa 3 weeks after rodenticide applications	There were no signs of rats or wild deer mice on the islands after poison applications. Deer mice that had been captured were released back onto rat-free islands 5 months after applications. In both years, >90% of the deer mice taken into captivity were released. Captive holding and translocation significantly reduced raptor mortality. One granivorous bird species, rufous-crowned sparrow, *Aimophila ruficeps* *Obscura* ,showed an unexpected significant decline	This was the first recorded rodent eradication that ensured a native endemic rodent, which showed to be equally susceptible to the bait as the rats, to survive. Eradication showed the importance of learning from previous operations, particularly based on species similar to raptors, as some granivorous birds may require captive-holding efforts or no-drop zones to minimize risk for non-target impacts as seen on Codfish Is, NZ. Demonstrates the need for well-designed data-driven mitigations.
Galapagos 2012 [27]	Pinzón giant tortoise (*Chelonoidis duncanensis*), Pinzón lava lizards (*Microlophus duncanensis*), Galapagos hawks (*Buteo galapagoensis*)	15 tortoises captured and held on another island for 2 years 40 lizards held in captivity until 10 days after second bait spread 60 hawks were captured and held in captivity until 12–14 days after second bait spread	All tortoises survived, were released, and have since bred 87% survival rate of lizards in captivity Unfortunately, 22 hawks died 12 to 170 days after the release of secondary poisoning therefore 10 were recapture treated with Vit K and not released until poison levels known to reduce	Rodenticide lasted longer in the environment than expected. Lizards did not eat bait in laboratory trials, but did in the field, emphasising both laboratory and field trials should be undertaken to determine species at risk
Lord Howe Island, Australia 2019 [28,29]	Lord Howe woodhen (*Gallirallus sylvestris*) and pied currawong (*Strepera graculina crissalis*)	Trial preemptive capture of both species prior to poison spread. 85% of woodhen population and 50% of currawong population captured before eradication operation and held until one month after.	All woodhen and currawong survived capitivity and woodhen population now quadruple pre-eradication population size	Importance of pre-emptive capture trials to understand how to manage wildlife in captivity Eradication also showed the importance of learning from previous operations particularly based on similar species
Gough Island, Tristan da Cunha, UK 2021 [30,31,32,33,34]	Gough bunting (*Rowettia goughensis*) and moorhen (*Gallinula comeri*)	Bunting and moorhens were trial preemptively captured and held before poison spread 84 moorhens and 100 buntings captured and held during poisoning	80 moorhens and 103 bunting released Buntings continue to do well; however, the status of moorhens is unknown Unfortunately, the rodent eradication was not a success	Recommended that the avicultural project be run separately but parallel to the eradication operation A dedicated husbandry team with a comprehensive plan was essential

## Data Availability

All data used in this review is already publicly available.

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
