# Peer review of "Lessons Learned for Pre-Emptive Capture Management as a Tool for Wildlife Conservation during Oil Spills and Eradication Events"

_animals, 2023, doi:10.3390/ani13050833_

Round 1
Reviewer 1 Report
Comments on Chilvers & McClelland
TITLE: Basically, the paper is a review of published examples of some actions to mitigate impacts on birds after oil-spills and actions to protect non-target animals before aerial baiting with brodifacoum against rodents. So the title needs to reflect this, e.g.,
A review of pre-emptive management of wildlife at risk from oil spills and from non-target poisoning during rodent eradications
LOGIC:
1. The review is not about all actions in an oil spill (i.e. it is not about rehabilitation of oiled animals per se, unless the rehabilitated animals are then held in captivity or translocated to a safe place along with other animals at risk or potential risk as the oil moves). This sort of limits the scope for the oily section to either in situ captivity until the oil goes away, or translocation far away for species that are unlikely to fly home before the oil goes away (penguins).
2. It seems to me a logical flow particularly for the non-target section on toxins would be to:
(a) Note which species on the islands were predicted to be at risk
(b) Describe the pre-emptive mitigation undertaken for each (in situ or ex situ captivity or translocation into the wild in a new place. Note: if individuals were poisoned, they might be rehabilitated with Vit K as an analogue of cleaning oiled birds. I know of no cases where this was done. But as with the rehab of oiled birds this is out of scope.
(c) Note whether this mitigation was actually necessary – did the brodifacoum kill any, some, all of those remaining at risk?. For iconic or rare species maybe any deaths would be unacceptable even if no risk was identified – tortoises are not really at risk to brodifacoum but had to be translocated as a PR exercise, and to set up a captive breeding option? . Did the populations recover (ex rats) irrespective of the mitigations. Could this be predicted, and the costs of mitigation avoided?
(d) Note whether the captive or translocated animals were returned to their island.
I attach a rough table I did to try and clarify for myself the ‘taxonomy’ of the process of pre-emption.
Oil spill: pre-emption is taking birds from oil-free areas, oil-free birds from the risk area, or rehabilitated oiled birds to in situ, ex-situ captivity, or translocate and release afar
Toxins: pre-emption is taking animals from the area prior to baiting into in situ, ex-situ captivity or translocated to new area and released.
3. CASE STUDIES:
These are a bit compact and hard to get to the point of the paper. I would have mini headings thus:
Background: A brief summary of the threat - some at the moment have too much detail
Species predicted to be at risk and capable of pre-emptive management: how was risk decided? Why are some managed and others not – practicality?
Pre-emptive strategy used: In-situ captivity, ex situ captivity, translocation to areas with or without conspecifics
Fate of translocated animals: Were the captive or translocated animals returned to the area (by themselves or with human help)?
Current status of the species:
Lessons: Which were actually at risk, fate of those left behind
4. INTRODUCTION
Too long and with extraneous bits.
Holding animals captive and/or translocation for re-release once a threat had been mitigated is a subset of the much wider use of these methods in wildlife management. Here we review … has been removed. Lines 47 onwards to 116 need to be severely pruned. – cut to the chase!
Some points may be better suited to the discussion – sea otters, dotterels for oil spills
4. SOME POINTS
Line 17: …oil spills or before aerial baiting (all case studies used brodifacoum as the toxicant) to eradicate introduced rodents are very …
Line 36: ‘exponential’ has a technical meaning and I doubt more cases meets it.
Line 42: ‘ wildlife in reaction to oil spills or before the use of poisons…’
Line 43: ‘drops’ is jargon. If you mean aerial baiting (all with brodifacoum) say so.
Line 44: the case studies have both translocation and captive holding options
Line 124: … pest eradication using toxicants on islands.
Line 155: Goldsworthy et al in a paper in this series list 10000 to 20000 penguins killed
Line 173: Goldsworthy et al might have recommended some trails to identify safe distances (depending on time of oil clean-up) for translocated penguins – but this seems impractical as it must be so case/species/site specific. In any case it is not their lesson you need to report but your lesson from their work – wording ‘it was’ to ‘it is’?
Line 178: 19000 penguins and yet endangered? A presume the species has some listing to justify this?
Line 288: Talon is a trade name, better to refer to the toxin used – brodifacoum.
Line 289: Odd that the non-native weka and kiwi were the focus. WSeka now almost considered a pest on Kapiti. Other species were included in the non-toxic trial, but not robins. Robins ‘previously identified as being at risk – I presume from precedents using brodifacoum, or more likely from 1080 cereal baiting?
Line 316: Gallirallus australis scottii – I think they were the Stewart Island subspecies?
Line 367 and 386: reptiles including tortoises are not affected by brodifacoum (see the reports on feeding trials with the toxin by Fisher reported in Rueda et al 2016). So, I think the translocation of the animals was more to assuage public perceptions than any real biological threat?
Line 405-414: The general point is the uncertainty in how long the toxin remains in the food chain (cf the sea and beach remain oily). I do not believe the key factor in persistence was aridity (plenty of other dry islands with no such problem). Rather a food-chain loop of toxin into invertebrates and/or lava lizards then not metabolised in the lava lizards which are prey for the hawks?
Line 445: logic – of course birds have to be taken in to captivity before the poison is dropped. The issue is how long before so the ‘dedicated husbandry teams’ recommended for other cases can adapt their husbandry.
Line 462: the threat to the moorhen was noted in Parkes (1996). Mice form a significant part of moorhen diet (they scavenge and hunt mice). Maybe no mice = less moorhen food = lower breeding success?
Line 495: We await the review with interest. The slug hypothesis is but one potential cause of the failure and , in my view, not the most likely.
5. DISCUSSION
Line 505: One more is the in-situ captivity of deermice during the rat eradication on Anacapa (Howard et al. 2009). Oryx 44: 30-40. Also ex-situ captivity of some raptors.
Line 514: .. generic?? … and constrained by practicality. This is where the pre-emptive strategy differs form the more usual rehabilitiation work done during oil spills. The former is predicated by a biological goal to mitigate the effects and restore the original population, the latter is often more aligned with animal welfare concerns – thus the extraordinary efforts to save one or two victims even when the overall population is abundant.
Line 528: There is the extensive planning for oil spills and its subset of ‘rat spills’ developed by the Alaskan government. I have seen the rat-spill component (Ebert et al. 2007). Bit old but I assume there are updated versions.
Line 555: Worth noting that some island projects went ahead with no mitigation despite clear risks to non-target wildlife – Macquarie, Hawadax, Antipodes, South Georgia etc. Why? No threat at the population or species level? No sensible mitigation possible?
REFERENCES
I have checked a few:
Line 646: Wegmann AND Howald have only their first initial used in the paper
Line 652: 2000 not 1999
Of all the species noted in the case studies, which ones were essential for pre-emptive mitigation strategies and which would have got by without? Selecting candidates for mitigation for oil spills seems more based on practicality than risk – thus penguins which can be captured and moved rather than most other seabirds which take what they get with a little rehabilitation to salve our welfare conscious. Selecting candidates for mitigation of risk due to primary or secondary poisoning during pest baiting should be based on risk. Thus the above point.
The paper touches on how such risks may be predicted but given the number of case species that proved not to be at risk maybe we need some better guidelines on how to assess matters.
Author Response
Dear Reviewer 1
Thank you for your review - Hopefully my replies to your comments are attached in a file and track changes can be seen in the manuscript. Your review has improved the manuscript. Thank you
Louise and Pete

Reviewer 2 Report
Dear authors, thank you for this interesting paper. This was one of the best papers I have ever revised for any scientific journal. Congratulations. I have made a few suggestions to add more details, especially on the methods.
Best regards.

Author Response
Dear Reviewer 2,
Thank you for your review and kind words.
Ln 97 Omnivores has been written in full
Ln 122 & Ln 131 Dates and clarification that PRISMA guidelines were used for this manuscript have been added in the first sentence of the methods "An online literature search was undertaken aligned with the PRISMA 2020 guidelines (14) with the aim of creating a list of publicly available articles or reports on the use of pre-emptive capture during oil spill response or island eradication, from 1970 to 2022.
Ln 263 The reference has been changed to a number - thank you sorry we missed it
Ln 285 A space has been entered in between Km2 and Kapiti
Ln 443 commas have been added where suggested
Ln 467 80 has been written out in full to Eighty
Ln 539 - 541 Sentences added to emphasize the point that lessons can only be learned if the information is written up and made available to learn from. The sentence added "However, this type of information needs to be written up and made publicly available from past responses for these lessons to be learned and used in the future."